# Bioavailability of Epigallocatechin Gallate Administered with Different Nutritional Strategies in Healthy Volunteers

**DOI:** 10.3390/antiox9050440

**Published:** 2020-05-19

**Authors:** Vicente Andreu Fernández, Laura Almeida Toledano, Nieves Pizarro Lozano, Elisabet Navarro Tapia, María Dolores Gómez Roig, Rafael De la Torre Fornell, Óscar García Algar

**Affiliations:** 1Grup de Recerca Infancia i Entorn (GRIE), Institut d’investigacions Biomèdiques August Pi i Sunyer (IDIBAPS), 08036 Barcelona, Spain; elnavarro@clinic.cat; 2Department of Nutrition and Health, Valencian International University (VIU), 46002 Valencia, Spain; 3Institut de Recerca Sant Joan de Déu (IR-SJD), 08950 Barcelona, Spain; lalmeida@sjdhospitalbarcelona.org (L.A.T.); lgomezroig@sjdhospitalbarcelona.org (M.D.G.R.); 4Maternal and Child Health and Development Network II (SAMID II), Instituto de Salud Carlos III (ISCIII), 28029 Madrid, Spain; 5BCNatal—Barcelona Center for Maternal Fetal and Neonatal Medicine, Hospital Sant Joan de Déu, Esplugues de Llobregat, 08950 Barcelona, Spain; 6Integrative Pharmacology and Systems Neurosciences Research Group, IMIM-Hospital del Mar Medical Research Institute, 08003 Barcelona, Spain; npizarro@imim.es (N.P.L.); rtorre@imim.es (R.D.l.T.F.); 7Department of Pharmacology, Therapeutics and Toxicology, Autonomous University of Barcelona, 08193 Bellaterra, Spain; 8CIBER of Physiopathology of Obesity and Nutrition (CIBEROBN), University Pompeu Fabra (CEXS-UPF), 08002 Barcelona, Spain; 9Department of Neonatology, Hospital Clínic-Maternitat, ICGON, IDIBAPS, BCNatal, 08028 Barcelona, Spain

**Keywords:** epigallocatechin gallate, EGCG, catechins, polyphenols, green tea, Teavigo^®^, bioavailability, pharmacokinetic profile, antioxidants, food supplement

## Abstract

The flavanol epigallocatechin gallate (EGCG) is being tested for the treatment of several diseases in humans. However, its bioavailability and pharmacokinetic profile needs a better understanding to enable its use in clinical trials. There is no consensus on the most appropriate concentration of EGCG in the body to obtain the maximum therapeutic effects. Therefore, the aim of this study is to analyze the bioavailability of EGCG orally administered alone or with different food supplements after overnight fasting in order to determine its optimal conditions (high concentrations in blood and the lowest interindividual variations) to be used as a pharmacological tool in human trials. Ten healthy volunteers (5 men and 5 women) aged 25 to 35 years were recruited prospectively. Three series of clinical experiments with a washout period of seven days among each were performed: (1) Teavigo^®^ (EGCG extract) alone, (2) Teavigo^®^ with a standard breakfast, and (3) FontUp^®^ (Teavigo^®^ commercially prepared with fats, carbohydrates, proteins, vitamins, and minerals). Blood samples were collected at 0, 30, 60, 90, 120, 180, 240, and 360 min after EGCG intake. Free EGCG in plasma was measured using a liquid chromatography and mass spectrometry UPLC-ESI-MS/MS analytical method. The pharmacokinetic variables analyzed statistically were area under the curve (AUC_0–360_), C_max_, C_av_, C_min_, T_1/2_, and T_max_. EGCG (Teavigo^®^) alone was the group with higher AUC_0–360,_ C_max_, and C_av_ both in men (3.86 ± 4.11 µg/mL/kg/6 h; 5.95 ng/mL/kg; 2.96 ng/mL/kg) and women (3.33 ± 1.08 µg/mL/kg/6 h; 6.66 ng/mL/kg; 3.66 ng/mL). Moreover, FontUp^®^ was the group with the highest value of T_1/2_ both in men (192 ± 66 min) and women (133 ± 28 min). Teavigo^®^ intake after fasting overnight revealed the highest concentration of EGCG in plasma according to its pharmacokinetic profile, indicating that this is an excellent alternative of administration if the experimental design requires good absorption in the gastrointestinal tract. Moreover, EGCG taken along with food supplements (FontUp^®^) improved the stability of the molecule in the body, being the best choice if the experimental design wants to reduce interindividual variation.

## 1. Introduction

Tea is one of the most popular beverages consumed across the world. It is extracted from the unfermented leaves of *Camellia sinensis* and mainly produced in four varieties, white, green, oolong, and black, depending on the oxidation and fermentation techniques applied [1]. Daily intake of green tea provides several health benefits, such as anti-inflammatory, anticarcinogenic, antimicrobial, and antioxidant effects reducing the risk of various diseases [2]. The health benefits of green tea are mainly attributed to its antioxidant properties [3]. For that reason, green tea extracts have been evaluated in diseases associated with an increase of reactive oxygen species (ROS) and oxidative stress, such as cancer and cardiovascular diseases [4,5]. Moreover other molecular mechanisms like signaling pathways, the modulation of some enzyme activities, and several interactions with membrane receptors related to cognitive functioning and Alzheimer’s disease have also been associated to green tea components [6,7].

Most of the health-promoting effects of green tea are associated to its polyphenol content [8], particularly flavonoids. The main flavonoids in green tea, the catechins, make up to 30%–40% of the solid components of green tea. The major catechins in tea are epicatechin (EC), epicatechin-3-gallate (ECG), epigallocatechin (EGC), and epigallocatechin-3-gallate (EGCG). EGCG, the most abundant flavanol, represents approximately 59% of the total catechins [2]. Many of the beneficial properties of green tea are attributed to this compound so that, recently, EGCG has been raised as a potential therapeutic tool [9,10]. Some of these health effects are related to the antiproliferative role of EGCG by the interference of the intracellular signaling cascades, which inhibits cell growth at the G1 stage, triggering apoptosis. In this framework, EGCG has been proposed as a chemopreventative in cancer prophylaxis [11,12]. Other beneficial properties of EGCG are its metabolic effects reducing the risk of type 2 diabetes and its cardiovascular complications [13]; antimicrobial activity due to the damage produced in the bacterial cell membrane when catechins bind to the lipid bilayer, inhibiting the ability of the bacteria to bind to the host cells; and its role in prevention and reduction of viral infections. EGCG also exercises a protective role in neurodegenerative diseases like Alzheimer disease [2] or after a neural injury [14]. Additionally, a recent research has demonstrated that this flavonoid improves cognitive performance and adaptive functionality in individuals diagnosed with Down syndrome by modulating the overexpression of the dual specificity tyrosine phosphorylation regulated kinase 1A (Dyrk1A) [15]. This protein is encoded by the *DYRK1A* gene, involved in signaling pathways which regulate cell proliferation, neural plasticity, and neurogenesis [16].

EGCG disposition depends on the ADME (absorption, distribution, metabolism, and excretion) processes, reaching the plasmatic peak concentration at 90 min and being undetectable 24 h after its oral intake [17]. EGCG reaches the stomach after oral administration, where the acid pH favors the structural stability of this molecule [18]. Then, a fraction of EGCG is absorbed in the small intestine. However, the low concentrations of EGCG observed in peripheral blood are caused by the transit of a substantial fraction of EGCG from the small to large intestines, where this molecule is transformed by the enterocytes and catabolized by local microbiota, leading to the formation of up to eleven catechin ring-fission products in different species. The EGCG ring-fission metabolites produced by intestinal microbiota are present as free and conjugated forms in plasma. In vitro data suggested that EGCG forms could reach the brain parenchyma by crossing the blood–brain barrier (BBB). Once distributed throughout the brain, EGCG promotes neuritogenesis, showing an important role in suppressing neurodegenerative diseases [19]. Finally, the liver cells metabolize the remaining EGCG, transforming into methylated, sulfated, and glucuronide intermediates which will be further eliminated in urine [20]. 

Moreover, several factors affect to the stability of polyphenols and promote remarkable differences in the pharmacokinetic parameters of EGCG among individuals [17,21]. The concentration of EGCG is reduced by high pH (4 or higher) and high temperatures (20 °C or higher), which directly affect its structural stability, promoting the chemical degradation of EGCG [22]. Recent studies postulate that the oral bioavailability of EGCG is low in humans and decreases if accompanied by food [17,23]. In contrast, others studies conclude that the intake of EGCG with some specific nutrients such as fish oil (omega-3 fatty acids) [24], vitamins as ascorbic acid which reduce the oxidation of EGCG [25], and minerals as selenium or chrome [25,26] improves the EGCG bioavailability, enhancing its antioxidant activity. 

Due to its hydrophilic nature, EGCGs show homogeneity problems in lipid products, affecting not only its appearance but also its effectiveness. Structural modifications of EGCG via esterification with aliphatic molecules such as long-chain fatty acids have been performed to solve these problems, and they can serve as a useful tool in altering its physical properties [27]. Interestingly, these ester derivatives have shown higher antiviral and antioxidant properties than the parent EGCG molecule and enhance its cellular absorption in vivo [27,28,29]. Moreover, EGCG derivates were demonstrated to be more effective in neuroprotection that non-modified EGCG in a cellular model of Parkinson [30]. Thus, EGCG esters may be used in nutrition and cosmetics as lipophilic alternatives with no effects in its functional properties.

Human studies evaluating the bioavailability of green tea extracts purified up to 95% in EGCG in different conditions (i.e., after overnight fasting, with additional nutrients, or included into food supplements) are scarce [31]. For that reason, the pharmacokinetic profile of EGCG in humans as well as its bioavailability is still investigated [21,32,33]. Different studies in both animal models and humans show controversial results regarding the bioavailability and also the biological effects elicited by EGCG [31,34]. Teavigo^®^ capsules are a purified green tea extract with an EGCG content around 94% for which solubility and stability have been improved from previous products containing green tea extracts. Teavigo^®^ is also incorporated into nutritional supplements as FontUp^®^, which also contains vitamins (A, C, D, E, K, and B1), minerals (zinc, chrome, selenium, etc.), and fatty acids as omega-3 in order to get better stability and to reduce the potential degradation of the molecule after dilution in various liquids such as milk and water [35]. Therefore, Teavigo^®^ is frequently used in clinical trials because it has enabled the use of EGCG with very minor contributions of other green tea catechins. In clinical trials, EGCG concentrations range from 100 mg to 600 mg per day [36]. A daily ration of nutritional supplement usually contains between 200 to 400 mg of Teavigo^®^. Doses higher than 800 mg per day have shown toxic effects in cellular and animal models with no additional benefits [37]. The European Food Safety Agency (EFSA) reviewed recently the safety of EGCG and concluded that there is evidence from interventional clinical trials that intake of doses equal or above 800 mg EGCG/day taken as a food supplement have been shown to induce a statistically significant increase of serum transaminases in treated subjects compared to controls. Other side effects described in this report include dizziness, anemia, hypoglycemia, and kidney problems [38]. 

A randomized and crossover study was performed in order to evaluate the bioavailability of EGCG in healthy subjects. EGCG concentrations in plasma samples of healthy male and female volunteers were analyzed after the administration of green tea extract Teavigo^®^ in three different ways: (1) fasting conditions, (2) combined with a Mediterranean diet breakfast, and (3) in the form of the commercial dietary supplement Fontup^®^. Thus, our results will determine the effect of different nutrients on the absorption and the stability of EGCG.

## 2. Materials and Methods 

### 2.1. Participants and Selection Criteria

Twenty-two Caucasian healthy adult volunteers were prospectively recruited in the study. All participants were physically examined before the start of the study to confirm their healthy condition. In the first clinic visit, the anthropometric parameters, gender, age, weight, and height, were collected and body mass index (BMI) was calculated for all individuals.

Exclusion criteria for subjects were a BMI of ≤18 kg/m^2^ or ≥30 kg/m^2^; smoking; pregnancy; use of oral contraceptives; any intake of medication; functional food or dietary supplement (including green tea); and any known renal, hepatic, gastrointestinal, hematological, endocrinological, pulmonary, cardiovascular, or malignant disease. Eight volunteers met exclusion criteria. The remaining fourteen subjects were included in the study and randomized by gender under equal conditions.

Sample size was calculated using a two-sided contrast method with a significance level of α = 0.05% and a statistical power of 80% (β = 0.2), assuming 10% of missing values. An accuracy value of 40 and a variance of 1000 in the control group were estimated based on blood EGCG values published previously [39]. Finally ten participants, five men and five women aged between 25 and 35 years of age, were selected and included in the study, discarding four participants not included in this age range [17,40,41].

Written informed consent was obtained from all the participants before the start of the study. All the protocols performed in this study were approved by the local ethical committee (CEIC-FSJD: Comitè Ètic d’Investigació Clínica—Fundació Sant Joan de Déu, ref. 2018/PIC-11-18) and were conducted according to the Declaration of Helsinki principles. All methods performed in this study were in accordance with the relevant guidelines and regulations. 

### 2.2. Reagents and EGCG Preparations

Teavigo^®^ (94% EGCG, 150 mg of green tea extract, 60 capsules) was provided by Healthy Origins (Pittsburgh, USA). The food supplement FontUp^®^ (94% EGCG, 266 mg of concentrated green tea extract plus fats, carbohydrates, proteins, vitamins, and minerals in format sachet; see Table A1) was purchased from Grand Fontaine Laboratories (Barcelona, Spain). The concentration (purity) of the EGCG of each product was stated by the manufacturers. 

Acetonitrile high-performance liquid chromatography (HPLC-grade), methanol (HPLC-grade), acetone (HPLC-grade), and glacial acetic acid (99.8%) of analytical grade (Scharlab, Barcelona, Spain) were used. Ultrapure water was obtained from a Milli-Q water purification system (Millipore, Bedford, MA, USA). Standards of EGCG and ethyl gallate (internal standard, I.S.), ascorbic acid (A92902) and Ethylenediaminetetraacetic acid (EDTA, E9884) were purchased from Sigma Aldrich (St. Louis, MO, USA). EGCG-free plasma for ultra-performance liquid chromatography-tandem mass spectrometry (UPLC-MS/MS) analysis was obtained from the Hospital del Mar blood bank (Barcelona, Spain). Monosodium dihydrogen orthophosphate (NaH_2_PO_4_:H_2_O) was purchased from Merck. All food ingredients for breakfast were purchased from local supermarkets.

### 2.3. Interventions

The clinical design included the administration of Teavigo^®^, Teavigo^®^ plus breakfast, and FontUp^®^ intake, with a washout period of seven days between interventions. Before starting the crossover study, the sequence of interventions was randomized for each subject. In all cases, the administration was performed after a fasting period of at least 8 h. No subject dropped out of the study.

For Teavigo^®^ series, the volunteers ingested 300 mg of green tea extract (two capsules with 140 mg of EGCG) with 100 mL of water. FontUp^®^ is a nutritional supplement with a cocoa taste formulated to provide the nutrients contained in a complete diet (188 kcal per ration, see Table A1) considering several comorbidities of subjects having Down syndrome (celiac disease, gluten intolerance, gastrointestinal disturbances, etc.). FontUp^®^ (266 mg of green tea extract, minimum 250 mg of EGCG) was dissolved into 200 mL of semi-skimmed milk (200 kcal) and administered with no additional breakfast. Moreover, in the Teavigo^®^ plus breakfast series, two capsules of Teavigo^®^ were administered with 30 g of breakfast cereals, 200 mL of semi-skimmed milk with soluble cocoa, and a couple of toasts with 5 mL olive oil, resulting in 480 kcal of caloric intake (Table 1). EGCG preparations were administered to all subjects within 5 min, and no additional food was taken for 6 h. 

### 2.4. Blood Collection, Processing, and Plasma Storage Protocol 

An 18-gauge shielded IV catheter with injection port (1.3 × 33 mm) and three-way stopcock extension (Vasofix^®^ Safety, Braun, Kronberg, Germany) for serial extractions was placed in the median cubital vein of the antecubital fossa for all the participants and was left in for 6 h. 

Five mL of whole blood was collected into lithium heparin tubes (BD Biosciences, Madrid, Spain) before the intervention and at 30, 60, 90, 120, 180, 240, and 360 min after the ingestion of each EGCG preparation. Immediately after collection, the blood samples were maintained on ice, and within 10 min of blood collection, plasma was separated by centrifugation for 10 min at 1750× *g* at 4 °C. Plasma samples (0.350 mL) were stored in Eppendorf low binding tubes (Sarstedt 72.706.600, Nümbrecht, Germany) containing 20 µL of a preserving solution (1.38 g NaH_2_PO_4_·H_2_O, 5 g of ascorbic acid, and 25 mg EDTA in 25 mL milli-Q H_2_O, pH 3.8) and kept at −80 °C until further analysis.

### 2.5. Determination of Free EGCG in Plasma Samples

#### 2.5.1. Preparation of Standard Solutions

Standard EGCG solutions (in methanol at 0.1% HCOOH) were prepared the day before each analytical batch and stored in a dark-glass flask at −18°C to prevent degradation. Working solutions of 0.1, 1, and 10 µg/mL were used to prepare calibration curves each analysis day, which consisted of at least two replicas at seven different concentrations (10, 29, 43, 71, 100, 286, and 429 ng/mL).

#### 2.5.2. Sample Preparation

The evaluation of free EGCG concentration in plasma samples was performed by an extraction procedure described by Martí et al. (2010) with some modifications [42]. Briefly, a solid-phase extraction was done in OASIS hydrophilic-lipophilic-balanced (HLB) µElution Plates 30 mm (Waters, Milford, MA, USA) that were conditioned sequentially with 250 µL of methanol and 250 µL of 0.2% acetic acid. Before loading to the plate, plasma samples were mixed with 350 µL of phosphoric acid 4% and centrifuged for 10 min at 4 °C and 16,000× *g*. Later, plates were washed with 200 µL of Milli-Q water and 200 µL of 0.2% acetic acid, and finally, samples were eluted with 100 µL of acetone/acetic acid solution 2% (70:30, *v*/*v*). Five µL of the eluted solution was directly injected in the UPLC-MS/MS. 

#### 2.5.3. UPLC-ESI-MS/MS

The analytical quantification was performed using an Acquity UPLC system (Waters Associates, Milford, MA, USA) coupled to a triple quadrupole (QuattroPremier, Waters, Milford, USA) mass spectrometer provided with an orthogonal Z-spray-electrospray interface (ESI) (Waters Associates). The chromatographic separation was performed at 55 °C using an Acquity UPLC BEH C18 (100 mm × 3.0 mm i.d., 1.7 μm) column at 0.3 mL/min of flow rate. The mobile phase consisted of formic acid 0.1% (A) and acetonitrile with formic acid 0.1% (B), with the following gradient program: from 99.5% B (maintained 1 min.) to 50% B in 1 min. After 2 min, the gradient was back to initial conditions. Total time of chromatogram was 6 min. 

The mass spectrometry (MS) used nitrogen as drying and nebulizing gas. The desolvation gas flow was set at 1200 L/h, and the cone gas flow was set at 50 L/h. The selected capillary voltage was 0.4 kV in negative ionization mode. The nitrogen desolvation temperature was 450 °C, and the source temperature was set to 120 °C. The collision gas was argon at a flow of 0.21 mL/min. The detection of the analytes was performed by the selected reaction monitoring (SRM) method. Mass/charge values selected for identification of analytes were as follows: EGCG *m*/*z* 457 → 139, 169, and 305; collision energy (CE) 15 eV for all the transitions; and ethyl gallate, *m*/*z* 197 → 78, 124, and 169 and CE 30, 10, and 10 eV, respectively.

### 2.6. Statistical Analysis

Database management and statistical analysis of the pharmacokinetic variables and the anthropometric measurements were performed using SPSS v.22 (IBM, Armonk, NY, USA) and GraphPad software 6.0 (Prism, San Diego, USA). Descriptive statistics were performed using mean, Standard Deviation (SD), and error (Std. Error). T-test was used to compare the distribution of age and gender for the different groups, using the Holm–Sidak correction. The one-way ANOVA and the Bonferroni post hoc tests were used to determine differences between the mean values of the anthropometric and pharmacokinetic variables obtained for the three EGCG series. Statistical significance was set at *p* < 0.05 for all the analyses performed. All pharmacokinetic data were calculated and presented in accordance with internationally accepted and standardized methods [43,44]. The correlations were also analyzed using linear regression analysis (SPSS v.22). Pharmacokinetic analysis was performed in accordance with current industry guidance for orally administered pharmaceutical products [44].

The maximum concentration of EGCG from time 0 to 6 h was defined as C_max_, with T_max_ being the time required to reach the C_max_. The concentration of plasma EGCG at the end of the dosing interval was defined as C_min_, and the mean concentration during the dosing interval was defined as C_av_. The plasma EGCG elimination half-life (T_1/2_) was calculated based on the formula T_1/2_ = 0.693/Ke, where Ke is the slope of the logarithmically transformed (ln) linear regression of the plasma EGCG concentrations. The area under the curve (AUC_0–360_) analysis was determined using the linear trapezoidal rule from 0–6 h.

## 3. Results

### 3.1. Anthropometric Data of Participants and EGCG Administration

Five healthy men and women were selected from the initial twenty-two volunteers to evaluate the EGCG concentrations in plasma (Figure 1). Sample size was calculated with a significance level of α = 0.05% and β = 0.2, obtaining a minimum value of ten subjects to get statistical significance. Previous studies on green tea or EGCG in volunteers were performed with a similar sample size [17,40,41]. The anthropometric data of all participants (age, height, weight, and BMI averages distributed by gender) were measured at baseline of the clinical trial (Figure 1, details in Table 2). Statistical analysis (*t*-test) demonstrated that there was no significant difference in age (*p*-value = 0.57) between men and women. The age average for all participants recruited for this study was 29.7 ± 4. By contrast, height, weight, and BMI data showed expected differences related to gender. For men, averages were height, 176 ± 5 cm; weight, 73.8 ± 11.6 kg; and BMI, 23.7 ± 2.8 (kg/m^2^). For women, the averages obtained were height, 163 ± 6 cm; weight, 52.8 ± 7.0 kg; and BMI, 19.9 ± 2.1 (kg/m^2^).

Three series of EGCG delivery randomized for each subject at the beginning of the trial were performed as follows (Figure 1): Teavigo^®^ in fasting conditions (without breakfast), administrating two capsules of Teavigo^®^ with a standard breakfast (480 kcal, see Table 1), and finally a sachet of FontUp^®^ (188 kcal, see Table A1) with semi-skimmed milk (200 kcal) containing at least 250 mg of EGCG. The nutrients included in the breakfast and their kcal are shown in Table 1. All experiments started after at least eight hours of overnight fast.

The selected dosage (300 mg of EGCG) has been previously tested in clinical trials for Down Syndrome [15,45], cardiovascular disease [46], and cancer [47,48], obtaining satisfactory results. In spite of the fact that the daily dose of EGCG can reach 800 mg per day, high doses are frequently administered two times per day: in the morning after overnight fasting and in the evening. Our experimental design has focused on the bioavailability of EGCG in the first daily dose in the morning used in clinical trials (between 200 and 400 mg of EGCG) performing different controlled conditions. Teavigo^®^ intake with no additional nutrients was used as the ideal condition for EGCG administration with the least interference from other environmental or nutritional factors.

### 3.2. EGCG Concentration–Time Profiles in Plasma 

The plasma EGCG concentrations were measured using ultra-performance liquid chromatography-electrospray tandem mass spectrometry (UPLC-ESI-MS/MS). This methodology has been previously validated [23,42]. Figure 2 shows analytical chromatograms of control (plasma at baseline) (Figure 2a), a blank spiked with a known concentration of an analytical standard of EGCG (Figure 2b), and a real sample (Figure 2c). No interfering peaks at the retention time of the EGCG peak were observed. 

The EGCG plasma concentrations over time considering the three interventions are shown in Figure 3 (individualized following gender) and in Figure 4 (mean values per gender). Concentration values were expressed as ng/mL of EGCG in plasma per kilogram of weight.

The delivery of EGCG in fasting conditions, taking two capsules of Teavigo^®^ without breakfast, showed higher concentrations of EGCG in plasma (C_max_ 5.9 for men and 6.7 ng/mL/kg for women) than Teavigo^®^ plus breakfast (C_max_ 3.9 for men and 4.5 ng/mL/kg for women) and five-folds higher than the administration of the food supplement FontUp^®^ containing EGCG (C_max_ 0.9 for men and 1.3 ng/mL/kg for women), being this difference statistically significant (*p*-value = 0.009 for men and 0.006 for women) (Figure 4, Table 3). Minimum concentrations, C_min_, after six hours of EGCG intake did not result in statistical differences between Teavigo^®^ and Teavigo^®^ plus breakfast. In contrast, Teavigo^®^ versus FontUp^®^ showed significant differences (Table 3). Interestingly, a slightly and nonsignificant trend was observed when differences between men and women were analyzed for C_max_ and C_min_, observing that the values of EGCG for all women included in this study were always higher than men (Figure A1). Moreover, the concentration–time curves showed a high variability among participants for Teavigo^®^ without breakfast with non-dependency by gender (Figure 3 and Figure 4, Table 3). This variability for EGCG concentration was also observed in the delivery of Teavigo^®^ plus breakfast. By contrast, the administration of FontUp^®^ generated more predictable concentrations in both men and women than the other treatments (Figure 3 and Figure 4, Table 3, and Table A2).

### 3.3. Pharmacokinetic Parameters of EGCG in Plasma 

The resulting values of the EGCG preparations showed differences by gender in the EGCG bioavailability (Figure 4). The peak of EGCG concentrations in plasma for Teavigo^®^ plus breakfast and FontUp^®^ intake reached later in women, with the T_max_ at 180 and 120 min respectively, than in men with the EGCG peak at 120 and 90 min. Conversely, the values of T_max_ for Teavigo^®^ administration were 120 min in men and 90 min in women (Table 3). These differences were analyzed by statistically using one-way ANOVA corrected by the Tukey test for multiple comparisons. Interestingly, only Teavigo^®^ plus breakfast versus Teavigo^®^ in women had statistical significance (*p*-value = 0.020).

Following a single-compartment modeling, the mean half-life averages (T_1/2_) were for men and women, respectively, Teavigo^®^, 154.0 ± 27.9 and 117.2 ± 53.5 min; Teavigo^®^ plus breakfast, 93.1 ± 36.2 and 111.4 ± 39.1 min; and FontUp^®^, 191.7 ± 66.4 and 132.9 ± 27.9. T_1/2_ showed statistical significance (*p*-value = 0.020) for Teavigo^®^ versus FontUp^®^ administration in men. These results highlight EGCG administration as FontUp^®^ reaches the longest half-life in the body, indicating that EGCG is more stable when is accompanied by vitamins (A, C, D, E, and K), folic acid, the omega-3 fatty acid, and the minerals present in FontUp^®^ than EGCG administered on an empty stomach or with a standard breakfast (Figure 4 and Figure 5). 

Statistical differences were also observed when C_av_ (ng/mL/kg) was analyzed in Teavigo^®^ versus FontUp^®^ preparations (Table 3, *p*-value = 0.025 for man and 0.005 for women). However, no statistical significance was observed between Teavigo^®^ and Teavigo^®^ plus breakfast for C_av_ (*p*-value = 0.34 for men and 0.23 for women, respectively). Teavigo^®^ administration showed a C_av_ of 3.0 ng/mL/kg for men and 3.7 ng/mL/kg for women, while Teavigo^®^ plus breakfast results were 1.5 ng/mL/kg for men and 2.1 ng/mL/kg for women. Additionally, the average concentration of EGCG contained in FontUp^®^ reached 0.6 ng/mL/kg for men and 0.9 ng/mL/kg for women. Considering that the values for all volunteers were corrected by their own body weight, these results showed nonsignificant differences by gender despite the amount of EGCG detected in the circulatory system being higher in women than men for all preparations tested (see Figure A1). 

One-way ANOVA analyses with Tukey post hoc test for multiple comparisons were performed to evaluate AUC_0–360_ between interventions. AUC_0–360_ for EGCG taken as Teavigo^®^ capsules in fasting conditions (Table 3) was significantly higher than the AUC_0–360_ for EGCG delivery with FontUp^®^ (*p*-value = 0.036 for men and 0.001 for women). Moreover, the difference between the AUC_0–360_ for Teavigo^®^ with breakfast and the AUC_0–360_ for EGCG present in FontUp^®^ was also statistically significant (*p*-value = 0.030 for men and 0.023 for women) (Table 3). These results indicated that EGCG delivery in FontUp^®^ supplement is the least efficient way to absorb EGCG to the body, compared with Teavigo^®^ with or without breakfast.

## 4. Discussion

In this study, we corroborated previous results regarding EGCG bioavailability depending on its ingestion with or without food [33]. The present study shows a higher bioavailability for the Teavigo^®^ intake with no additional food compared to the other two preparations tested taken either with food or with a dietetic supplement. The increased global bioavailability of Teavigo^®^ without food shows also a higher interindividual variability (up to 100% in males, based on AUC values). The FontUp^®^ product displays the lowest bioavailability, less variability, and higher stability on EGCG plasma concentrations.

EGCG and other green tea polyphenols may have potential therapeutic applications mainly in the prevention of a large variety of human diseases. A large number of clinical studies have demonstrated the benefits of EGCG in patients diagnosed with cancer [49], neurodegenerative diseases [50], Down syndrome [51], and metabolic syndrome [52] by regulating various metabolic, genetic, and epigenetic pathways [53,54]. Several studies have shown important differences regarding the bioavailability of this molecule due to the heterogeneity of the human populations analyzed [21,55] and the differences in stability depending on the delivery strategy [23,35,39]. Additionally, the absorption and stability of EGCG are directly influenced by the combined intake of this molecule with other food products, which determine the environment of EGCG before its absorption and modulates its biological response. Unfortunately, the interactions of specific nutrients with the metabolic processes related to EGCG bioavailability remain unclear [23,31].

EGCG has been tested for therapeutic purposes in a range of doses from 150 to 400 mg per day, being the oral administration the most widely used delivery method in animal models and clinical trials in humans [31,45,56]. The blood concentrations obtained in this study for Teavigo^®^ alone or with a Mediterranean diet breakfast (Table 3 and Table A2) are consistent with previous works which analyzed the bioavailability of EGCG in a limited number of volunteers [23,36,55] or in a specific condition [21,39]. The present study moves one step forward using different nutritional strategies to deliver EGCG as well as a cohort of volunteers with an adequate sample size to analyze the EGCG dose used in clinical trials [15], taking also into account the gender balance. The objective is to determine the most appropriate conditions for EGCG intake.

EGCG has been administered in the form of green tea extract (Teavigo^®^), in a single dose of 250 mg after overnight fasting, showing significant differences according to the conditions and nutritional supplements used [23]. The intake of Teavigo^®^ after fasting overnight results in the highest peak concentrations (C_max_) and AUC_0–360_ values in both genders (Figure 5). The comparison between Teavigo^®^ and Teavigo^®^ plus breakfast for each participant indicates that a Mediterranean diet breakfast reduces the bioavailability of EGCG (more than 100% in males and 30% in females following AUCs). For that, the green tea extract should be ingested alone after overnight fasting to optimize the gastrointestinal absorption of the EGCG. These results are consistent with previous studies in which the authors proposed that the administration of EGCG alone elicits an attenuated strong response from the stomach and pancreas, minimizing the digestion processes [17,57]. When a capsule containing EGCG with no additional food is ingested after fasting overnight and arrives to the stomach, the gastrointestinal processes are only partially activated just by that small amount of nutrients. Then, the EGCG molecules remain stable due to the propitious acidic environment (pH < 3), where the oxidation of their polar residues is minimized [58]. In addition, the neutralization process by the secretion of bile salts is not activated, which favors and increases their absorption by the enterocytes in the small intestine. Otherwise, the lack of activation of digestion mechanisms reduces the activity of the bacterial microbiota responsible for catabolizing these antioxidant molecules in the large intestine. 

As mentioned before, a standard breakfast composed by milk, cocoa, cereals, and toasts with olive oil reduces the bioavailability of EGCG (Figure 5), in accordance with results obtained in other studies [23,35,39]. For example, some authors have concluded that the combined intake of Teavigo^®^ with semi-skimmed milk generates a significant decrease of EGCG bioavailability [23,35,39]. Moreover, Teavigo^®^ taken with breakfast and, to a lesser extent, FontUp^®^ showed a delay (T_max_) to reach the higher concentration peak (C_max_) of bioavailability in plasma (Table 3 and Figure 3, Figure 4 and Figure 5). The intake of food products delays the rate of gastric emptying, which is critical to determine the absorption rate in the small intestine and influences the bioavailability of orally administered drugs [56]. More likely, the intake of enough amount of nutrients with EGCG activates the gastrointestinal processes and delays the gastric emptying. Consequently, the EGCG molecules spend more time on the basic environment of the small intestine, increasing their degradation. Therefore, the absorption of EGCG with additional nutrients is not only reduced but also delayed as compared to EGCG alone, which is able to pass directly to small intestine to be absorbed.

We also addressed the role of specific nutrients on the bioavailability of EGCG when those are ingested together with this molecule. In this line, the administration of the food supplement FontUp^®^ has reported interesting results when it was compared with the intake of Teavigo^®^ plus breakfast. It has been previously demonstrated that EGCG is unstable in environments with high temperature [59] and basic pH [60]. Accordingly, all the experiments have been performed using fresh fluids at 4 °C to reduce the degradation of EGCG, water in the case of Teavigo^®^ alone, and semi-skimmed milk for the other two series of EGCG administration: FontUp^®^ and Teavigo^®^ plus breakfast. In reference to pH, when the food arrives to the small intestine, the pancreatic juice neutralizes the hydrochloric acid emptied into the duodenum from the stomach. Most likely, it is even more decisive that there is a lack of activation of gastrointestinal processes from the pH of the fluids and nutrients taken along with EGCG. 

Moreover, ascorbic acid has been used in previous studies as a preservative to improve the bioavailability of EGCG by preventing oxidation at an acidic pH [25,57,61]. Vitamin C keeps the polarity of the eight hydroxyl groups and the structural stability of the molecule. The ratio used in FontUp^®^ (12 mg of ascorbic acid per 250 mg of pure EGCG, see Table A1) reported the best stability parameters (T_1/2_ =163 min) after gastrointestinal absorption compared to Teavigo^®^ (T_1/2_ =135 min) after fasting and Teavigo^®^ plus a breakfast (T_1/2_ =102 min). However, the presence of vitamin C in FontUp^®^ did not improve the absorption of the molecule as did the presence of either sucrose previously described in literature [25]. Levels of EGCG in blood were 5-fold lower with FontUp^®^ (C_av_ = 0.74 ng/mL/kg) compared to Teavigo^®^ alone (C_av_ = 3.3 ng/mL/kg) and 2.5-fold lower than Teavigo^®^ plus breakfast (C_av_ = 1.8 ng/mL/kg).

Otherwise, omega-3 polyunsaturated fatty acids from fish oil has been also tested to improve the intestinal absorption of EGCG [24,26,62]; however, its presence in FontUp^®^ does not result in an improvement of the EGCG bioavailability compared to Teavigo^®^ in fasting conditions or after a standard breakfast.

New delivery technologies to improve the oral bioavailability of EGCG have been raised to expand its application in a lipophilic media. Esterification of the water-soluble EGCG with stearic acid (SA), eicosapentaenoic acid (EPA), and docosahexaenoic acid (DHA) or replacing the hydroxyls of EGCG with acetyl groups have demonstrated a significant improvement in stability, bioavailability, and bioactivities as antioxidants or antiproliferatives [27,30]. For that, their potential applications in future clinical studies should be analyzed in detail.

In reference to the limitations of our study, in spite of the sample size being calculated to obtain significant results in three series of experiments (*n* = 10), a higher number of volunteers would be desirable to reduce the interindividual variability related to EGCG disposition. Moreover, the main cause of interindividual variability is the low EGCG bioavailability estimated in humans (0.1%–0.3%) [63,64]. Small changes in this percentage result in significant changes in plasma concentrations beyond the differences among individuals in the genetic background related to EGCG metabolism. For example, a variation of EGCG bioavailability from 0.3% to 0.6% causes an increase of 100% in the EGCG plasma concentration. For these reasons, the present crossover study tries to reduce the impact of interindividual variation due to each volunteer used as its own control. 

The oral administration of a green tea extract with a high percentage of EGCG (Teavigo^®^) has been unequivocally demonstrated as the proper form to obtain the higher bioavailability values. In contrast, EGCG accompanied by specific nutrients inside the food supplement FontUp^®^ showed the most homogeneous disposition for all participants. Recently, it has been shown that FontUp^®^ at doses within the range of those used in clinical studies normalized brain and plasma biomarkers deregulated in Dryk1a transgenic mice (TgBACDyrk1A), without negative effects on liver and cardiac functions [65]. Therefore, FontUp^®^ or EGCG alone will be selected for clinical trials depending on the experimental design of the study, the goals, the variables to study, the environmental conditions, and other particular considerations.

## 5. Conclusions

EGCG is being evaluated as a promising compound for the treatment of human noncommunicable diseases such as cancer and cardiovascular, hepatic, and neurodegenerative diseases. However, the exposure to EGCG required for the treatment of each pathology is still under study. Additionally, it is well known that the EGCG bioavailability shows a high interindividual variability related to the gastrointestinal absorption, the stability of the molecule, the nutritional environment, and the administration conditions. The present research highlights that the use of a green tea extract enriched with EGCG (Teavigo^®^, 94% EGCG) after fasting overnight leads to the highest exposure to EGCG both in men and women (AUC_0–360_ and C_max_). Moreover, Teavigo^®^ ingested with a Mediterranean diet breakfast shows a reduction in EGCG bioavailability. However, neither are the nutrients ingested at breakfast decisive in blocking the absorption of ECGC nor do the contents of the nutritional supplement FontUp^®^ promote such absorption. In contrast, the supplements contained in FontUp^®^ are able to favor the stability of EGCG in the gastrointestinal tract, and therefore, its use may be appropriate when the experimental design needs to reduce the interindividual variability and to analyze the efficacy of a stable and similar EGCG concentration in all study participants. 

## Figures and Tables

**Figure 1 antioxidants-09-00440-f001:**
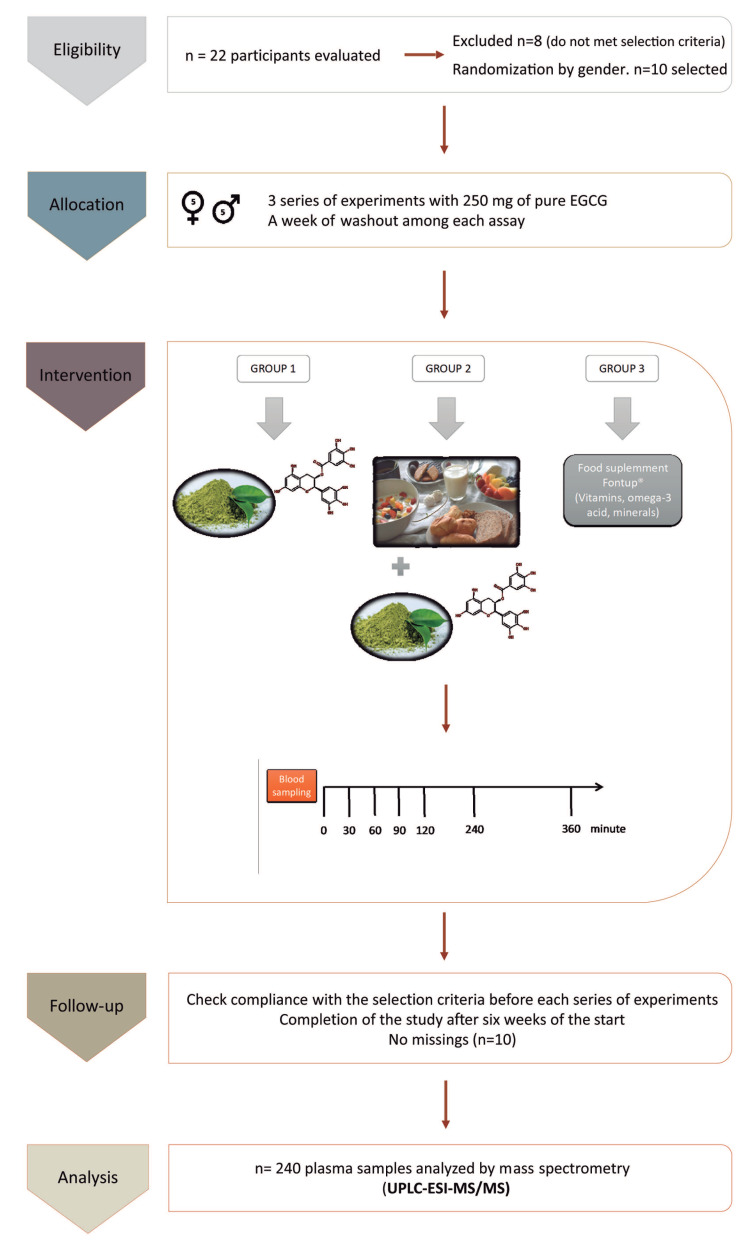
Consort flow chart and schematic representation of the study design and the blood sampling: The experimental design includes the study groups, the timeline of blood extractions after the intake of epigallocatechin-3-gallate (EGCG), and the methodology to analyze the samples. UPLC-ESI-MS/MS = ultra-performance liquid chromatography-electrospray tandem mass spectrometry.

**Figure 2 antioxidants-09-00440-f002:**
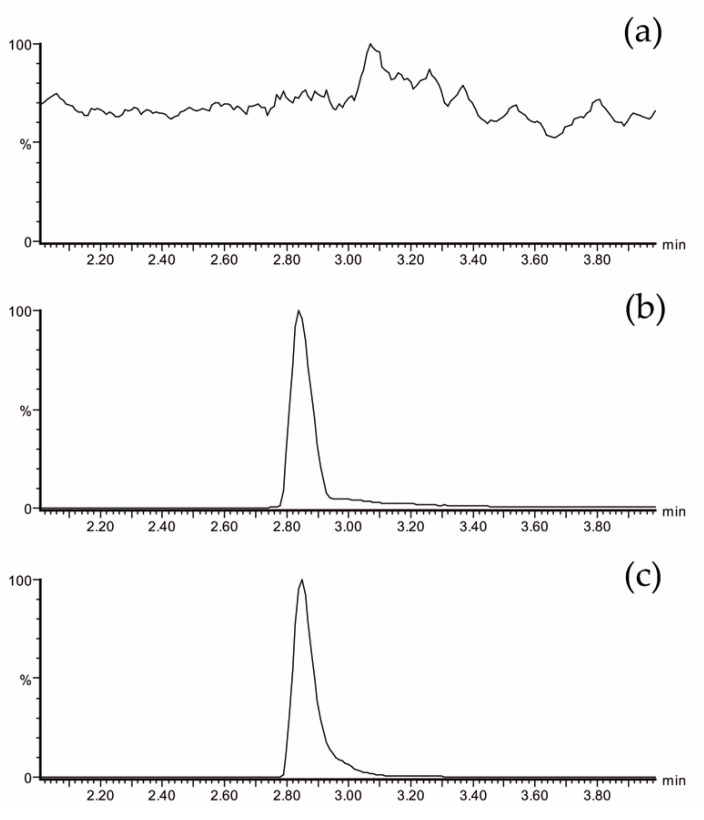
LC-MS/MS chromatogram of EGCG showing the most specific and sensitive MS/MS transition m/z 457→169: (**a**) plasma sample from a volunteer at baseline, (**b**) blank of plasma spiked with 428 ng/mL of EGCG, and (**c**) plasma sample from a volunteer at 60 min after Teavigo^®^ ingestion (calculated concentration: 289 ng/mL). LC-MS/MS = liquid chromatography tandem mass spectrometry.

**Figure 3 antioxidants-09-00440-f003:**
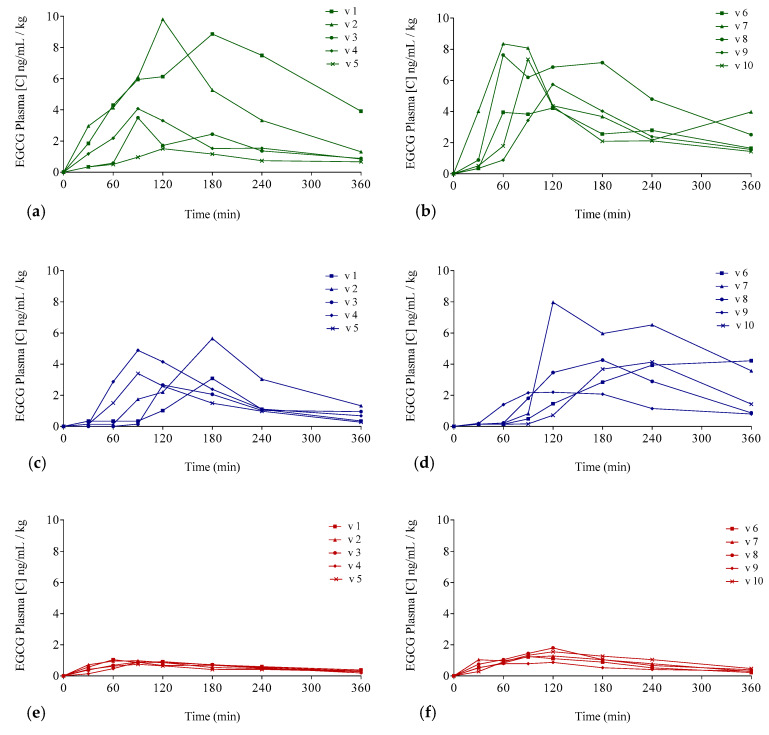
Plasma EGCG concentration–time curves for each of the participants for the three different EGCG preparations: Concentrations for men (**a**,**c**,**e**) and (**b**,**d**,**f**) for women. Teavigo^®^ condition is shown by green lines (**a**,**b**), Teavigo^®^ plus breakfast is shown by blue lines (**c**,**d**), and FontUp^®^ is shown by red lines (**e**,**f**).

**Figure 4 antioxidants-09-00440-f004:**
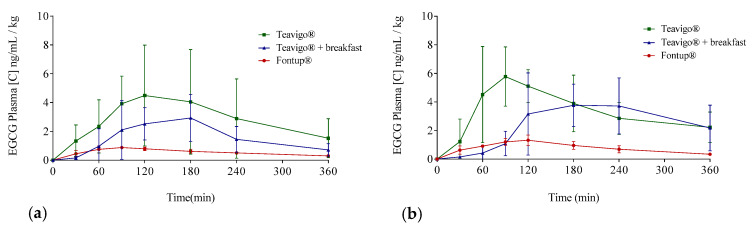
Averages of EGCG plasma concentrations over time distributed following EGCG preparation and gender: Teavigo^®^ condition is shown in green lines, Teavigo^®^ plus breakfast is in blue lines, and FontUp^®^ is in red lines. (**a**) EGCG average concentrations for men and (**b**) EGCG average concentrations for women. Error bars represent the standard deviation.

**Figure 5 antioxidants-09-00440-f005:**
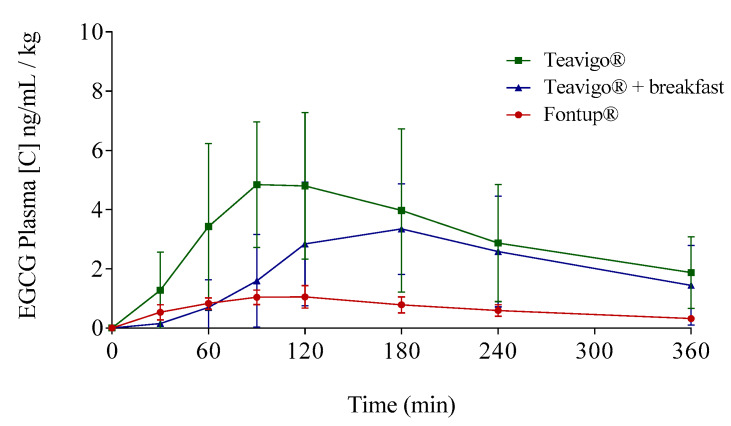
Mean plasma concentration time profiles of EGCG for the three different EGCG interventions by the oral route: Green line represents Teavigo^®^, blue line represents Teavigo^®^ plus breakfast, and red line represents FontUp^®^. Each value was obtained by HPLC-MS analysis. Error bars represent the standard deviation.

**Table 1 antioxidants-09-00440-t001:** Nutrients of the standardized breakfast included in the Teavigo^®^ plus breakfast intervention.

Nutrient	Weight (g)	% (*w*/*w*) ^1^	kcal
Semi-skimmed milk + Soluble cocoa	200	78.4	200
Breakfast cereals	30	11.8	120
2 bread toasts + 5 mL Olive oil	25	9.8	160
**Total**	**255**	**100**	**480**

^1^ w = weight.

**Table 2 antioxidants-09-00440-t002:** Anthropometric data of the participants included in the study.

Demographic Parameters	Total	Men	Women
Nº of participants	10	5	5
Age mean (years)	29.7 ± 4.3	30.6 ± 5.2	28.8 ± 3.6
Height (cm)	169 ± 8.8	176 ± 5.0	163 ± 6.0
Weight (kg)	63.2 ± 14.4	73.8 ± 11.6	52.8 ± 7.0
BMI ^1^ (kg/m^2^)	21.8 ± 3.1	23.7 ± 2.8	19.9 ± 2.1

^1^ BMI = body mass index.

**Table 3 antioxidants-09-00440-t003:** Plasma kinetic parameters for EGCG after the three preparations.

Parameters	G	Teavigo^®^	Teavigo^®^ with Breakfast	FontUp^®^
AUC_0–360_ (μg/mL/kg/6 h)	♂ ♀	3.9 ± 4.1 3.3 ± 1.1	1.5 ± 0.6 2.4 ± 1.1	0.6 ± 0.1 ** 0.8 ± 0.2 **
C_max_ (ng/mL/kg)	♂ ♀	5.9 ± 4.1 6.7 ± 1.7	3.9 ± 1.3 4.5 ± 2.1	0.9 ± 0.1 ** 1.3 ± 0.3 **
C_av_ (ng/mL/kg)	♂ ♀	3.0 ± 2.6 3.7 ± 2.3	1.5 ± 1.5 2.1 ± 2.0	0.6 ± 0.2 ** 0.9 ± 0.4 **
C_min_ (ng/mL/kg)	♂ ♀	1.5 ± 1.4 2.2 ± 1.1	0.7 ± 0.4 2.2 ± 1.6	0.3 ± 0.1 * 0.3 ± 0.0 *
T_1/2_ (min)	♂ ♀	154.2 ± 27.9 117.2 ± 53.5	93.1 ± 36.2 * 111.4 ± 39.1	191.7 ± 66.4 132.9 ± 27.9
T_max_ (min)	♂ ♀	120 (90–180) 90 (60–120)	120 (90–180) 180 (120–360)	90 (60–90) 120 (90–120)

Values are means ± standard deviations for the 10 participants (*n* = 10), except for T_max_ (median with the minimum and maximum time observed). * Value is significantly different from the other values in the row at the level of *p* < 0.05. ** Value is significantly different from the other values in the row at the level of *p* < 0.001. Abbreviations: G = gender; min = minutes; h = hours; ♂ = man; ♀ = women; AUC_0–360_ = area under the curve from 0 to 6 h; C_max_ = maximum concentration; C_av_ = average concentration; C_min_ = minimum concentration (at the end of the treatment); T_1/2_ = half-life; T_max_ = time required to reach the maximal concentration.

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
