# Peer review of "Bioavailability of Epigallocatechin Gallate Administered with Different Nutritional Strategies in Healthy Volunteers"

_antioxidants, 2020, doi:10.3390/antiox9050440_

Round 1

Reviewer 1 Report

In this manuscript authors report a study that analyze the bioavailability of EGCG orally administered alone or with various food supplements in order to determine its optimal conditions to be used in pharmacological tool in human trials. The topic in interesting as EGCG has gained much attention for its various beneficial health benefits.

In my opinion the paper can be accepted after minor revisions:
Authors should enlarge the introduction paragraph including some relevant and recent studies on the preparation and use of EGCG derivatives. In fact, EGCG can be modified structurally to improve its lipophilicity for application in lipophilic medium, and enhance its cellular absorption in vivo. Below are some recent papers:

• (2011) Advances in Clinical Chemistry, 53, 155– 177
• (2011) Journal of Agricultural Food Chemistry, 59, 6526–6533;
• (2012) Journal of Functional Foods, 4, 87-93.
• (2012) Food Chemistry, 131, 22–30.
• (2015) Neuroscience Letters, 429, 36-364;
• (2018) Natural Product Communications Vol. 13 (9) -1117-1122

The manuscripts is well written and it is easy to read.

in figure 1 “n=240 plasma…….mass espectrometry” should be changed to “n=240 plasma…….mass spectrometry”
page 8 line 277 authors claim “No interfering peaks at the retention time of the EGCG peak were observed” however figure 2 report a peak for ECGC that result too enlarged and therefore poor resolution. In addition they shoud report the image of the mass spectrum for ECGC and MS/MS analysis.

Author Response

We really appreciate your constructive comments. Thanks for that. We have included all in the reviewed version of the manuscript as follows:

- Authors should enlarge the introduction paragraph including some relevant and recent studies on the preparation and use of EGCG derivatives. In fact, EGCG can be modified structurally to improve its lipophilicity for application in lipophilic medium, and enhance its cellular absorption in vivo. Below are some recent papers:

As the referee suggests, introduction has been enlarge with additional information about the topic of EGCG derivatives and the papers suggested has been referenced. We have emphasized that the esterification of EGCG solves the problem of its use in lipophilic environments, improving the spectrum of use in different matrices. The paragraph was added as follows (lines 106-114):

“Due to its hydrophilic nature, EGCG show homogeneity problems in lipid products, affecting not only its appearance but also its effectiveness. Structural modifications of EGCG via esterification with aliphatic molecules such as long-chain fatty acids have been performed to solve these problems and they can serve as a useful tool in altering its physical properties [27]. Interestingly, these ester derivatives have shown higher antiviral and antioxidant properties than the parent EGCG molecule and enhance its cellular absorption in vivo [27-29]. Moreover, EGCG derivates demonstrated to be more effective in neuroprotection that non-modified EGCG in a cellular model of Parkinson [30]. Thus, EGCG esters may be used in nutrition and cosmetics as lipophilic alternatives with no effects in its functional properties.”

Furthermore, a reference to the cited studies appears in a new paragraph of the discussion, drawing attention to the use of these alternatives in future clinical studies (lines 499-505):

“New delivery technologies to improve the oral bioavailability of EGCG have raised to expand its application in a lipophilic media. Esterification of the water-soluble EGCG with stearic acid (SA), eicosapentaenoic acid (EPA), and docosahexaenoic acid (DHA) or replacing the hydroxyls of EGCG with acetyl groups have demonstrated a significant improvement in stability, bioavailability and bioactivities as antioxidant or antiproliferative [27,30]. For that their potential applications in future clinical studies should be analyzed in detail.”

Please, note that the reference ((2011) Advances in Clinical Chemistry, 53, 155– 177) has been inserted in the second paragraph of the introduction (line 72) due to its adequacy with the content. “In this framework EGCG is has been proposed as chemopreventive in the prophylaxis of cancer [11] [12].”

References added:

(2011) Advances in Clinical Chemistry, 53, 155– 177

(2011) Journal of Agricultural Food Chemistry, 59, 6526–6533;

(2012) Journal of Functional Foods, 4, 87-93.

(2012) Food Chemistry, 131, 22–30.

(2010) Neuroscience Letters, 429, 36-364.

- In figure 1 “n=240 plasma…….mass espectrometry” should be changed to “n=240 plasma…….mass spectrometry”

All grammatical errors have been fixed and English language and style has been checked by Native English.

page 8 line 277 authors claim “No interfering peaks at the retention time of the EGCG peak were observed” however figure 2 report a peak for ECGC that result too enlarged and therefore poor resolution. In addition they should report the image of the mass spectrum for ECGC and MS/MS analysis.

As reviewer suggests Figure 2 has been replaced with an improved figure with a better resolution. As for the reviewer’s comment that says that the figure should include EGCG mass spectrum and the MS/MS analysis, we want to clarify that chromatograms correspond, indeed, to a specific MS/MS transition, although it may be misinterpreted as a total ion chromatogram. In order to avoid any further misunderstanding, we have included a more accurate description in the figure footnote (Lines 317-321):

“Figure 2. LC-MS/MS chromatogram of EGCG showing the most specific and sensitive MS/MS transition m/z 457→169 (a) plasma sample from a volunteer at baseline, (b) blank of plasma spiked with 428 ng/mL of EGCG, and (c) plasma sample from a volunteer at 60 minutes after Teavigo® ingestion (calculated concentration: 289 ng/mL).”

Reviewer 2 Report

This study addresses the bioavailability assessment of epigallocatechin gallate (EGCG) administered in humans by distinct nutritional approaches. The authors conducted a very interesting and well-structured study, and the subject is certainly of interest to the readers of the journal. Facing the aforementioned, I would recommend the authors to perform some minor corrections and improvements before acceptance.

-Some sentences are too long, as lines 56-61, e.g.

-Grammatic flaws, misspelled words, and formatting problems exist over the manuscript. Check examples in lines 70, 78, 79, 82, 90, 94, Fig.1, etc.

-The authors should fully describe abbreviations for the first time those appear in the text.

-I would recommend also the authors better justify why they have considered Teavigo® samples for analysis, and also the selected dosage of these extracts. Moreover, information regarding the number of participants involved in the study just be also justified particularly regarding other similar studies.

-In addition, I would recommend that the authors include in the discussion some data and prospects related to new delivery technologies able to improve the oral bioavailability of EGCG as viable alternatives for these specific samples.  

Author Response

We really appreciate your constructive comments. Thanks for that. We have included all in the reviewed version of the manuscript as follows:

-Some sentences are too long, as lines 56-61, e.g.

Following reviewer suggestions, we modified the sentence (lines 54-62) as follow:

“Daily intake of green tea provides several health benefits, such as anti-inflammatory, anticarcinogenic, antimicrobial and antioxidant effects reducing the risk of various diseases [2]. The health benefits of green tea are mainly attributed to its antioxidant properties [3]. For that reason green tea extracts have been evaluated in diseases associated with an increase of reactive oxygen species (ROS) and oxidative stress, such as cancer and cardiovascular diseases [4,5]. Moreover other molecular mechanisms like signaling pathways, the modulation of some enzyme activities and several interactions with membrane receptors related to cognitive functioning and Alzheimer’s disease have also been associated to green tea components [6,7].”

-Grammatic flaws, misspelled words, and formatting problems exist over the manuscript. Check examples in lines 70, 78, 79, 82, 90, 94, Fig.1, etc.

All grammatical errors mentioned have been fixed and English language and style has been checked by Native English. For example:

Line 70 “In this framework EGCG is has been proposed as chemopreventive in the prophylaxis of cancer” has been changed by “In this framework EGCG has been proposed as a chemopreventive in cancer prophylaxis” (current line 71)

Line 78 “individuals having Down Syndrome, by modulating the overexpression of the protein kinase Dyrk1A” has been replaced “individuals diagnosed with Down syndrome by modulating the overexpression of the dual specificity tyrosine phosphorylation regulated kinase 1A (Dyrk1A)” (current line 79-80)

Line 79 “This protein is encoded by the gene DYRK1A, involved in signaling pathways regulating cell proliferation, neural plasticity and neurogenesis” has been changed by “This protein is encoded by DYRK1A gene, involved in signaling pathways which regulates cell proliferation, neural plasticity and neurogenesis” (current lines 80-82).

Line 82 “reaching the peak plasma concentrations at 90 minutes and being undetectable 24h its oral intake” has been corrected as follows “reaching the plasmatic peak concentration at 90 minutes and being undetectable 24h after its oral intake” (current lines 84-85)

Line 90 “and induce neuritogenesis, suggesting that they might be important in suppressing neurodegenerative diseases” has been replaced by“Once distributed throughout the brain EGCG promotes neuritogenesis, showing an important role in suppressing neurodegenerative diseases” (current lines 93-94)

Line 94 “Moreover, several factors affect to the stability of polyphenols promoting remarkable differences in the pharmacokinetic parameters of EGCG among individuals” has been corrected as follows “Moreover, several factors affect to the stability of polyphenols and promote remarkable differences in the pharmacokinetic parameters of EGCG among individuals” (current lines 97-98).

-The authors should fully describe abbreviations for the first time those appear in the text.

We checked all abbreviations and we fully described in the text. Some examples are:

              Line 80: dual specificity tyrosine phosphorylation regulated kinase 1A (Dyrk1A)

              Line 83: ADME (absorption, distribution, metabolism and excretion)

              Line 148: body mass index (BMI)

              Line 172: high-performance liquid chromatography (HPLC-grade)

              Line 177: ultra-performance liquid chromatography – tandem mass spectrometry (UPLC-MS/MS)

              Line 220: hydrophilic-lipophilic-balanced (HLB)

              Line 235: mass spectrometry (MS)  

              Line 313: ultra performance liquid chromatography-electrospray tandem mass spectrometry (UPLC-ESI-MS/MS)

-I would recommend also the authors better justify why they have considered Teavigo® samples for analysis, and also the selected dosage of these extracts. Moreover, information regarding the number of participants involved in the study just be also justified particularly regarding other similar studies.

As reviewer suggested, we have included some sentences in the results section in order to clarify the use of Teavigo® and the dosage (lines 282-289):

“The selected dosage (300mg of EGCG) has been previously tested in clinical trials for Down Syndrome [15,44], cardiovascular disease [45] and cancer [46,47] obtaining satisfactory results. In spite of the daily dose of EGCG can reach 800 mg per day, high doses are frequently administered two times per day: in the morning after overnight fasting and in the evening. Our experimental design has focused on the bioavailability of EGCG in the first daily dose in the morning used in clinical trials (between 200 and 400mg of EGCG) performing different controlled conditions. Teavigo® intake with no additional nutrients was used as ideal condition for EGCG administration with the least interference from other environmental or nutritional factors.”

  • De la Torre R, De Sola S, Pons M, et al. Epigallocatechin-3-gallate, a DYRK1A inhibitor, rescues cognitive deficits in Down syndrome mouse models and in humans. Mol Nutr Food Res. 2014;58(2):278‐288. doi:10.1002/mnfr.201300325.

  • de la Torre R, de Sola S, Hernandez G, et al. Safety and efficacy of cognitive training plus epigallocatechin-3-gallate in young adults with Down's syndrome (TESDAD): a double-blind, randomised, placebo-controlled, phase 2 trial. Lancet Neurol. 2016;15(8):801‐810. doi:10.1016/S1474-4422(16)30034-5.

  • Widlansky ME, Hamburg NM, Anter E, et al. Acute EGCG supplementation reverses endothelial dysfunction in patients with coronary artery disease. J Am Coll Nutr. 2007;26(2):95‐102. doi:10.1080/07315724.2007.10719590.

  • Dostal AM, Samavat H, Bedell S, et al. The safety of green tea extract supplementation in postmenopausal women at risk for breast cancer: results of the Minnesota Green Tea Trial. Food Chem Toxicol. 2015;83:26‐35. doi:10.1016/j.fct.2015.05.019

  • Kumar NB, Pow-Sang J, Egan KM, et al. Randomized, Placebo-Controlled Trial of Green Tea Catechins for Prostate Cancer Prevention. Cancer Prev Res (Phila). 2015;8(10):879‐887. doi:10.1158/1940-6207.CAPR-14-0324.

In reference of the sample size selected, we added the following sentences at the starting of results section (lines 265-269):

“Five healthy men and women were selected from initial twenty two volunteers to evaluate the EGCG concentrations in plasma (Figure 1). Sample size was calculated with a significance level of α = 0.05% and β = 0.2, obtaining a minimum value of ten subjects to get statistical significance. Previous studies on green tea or EGCG in volunteers were performed with a similar sample size [17,40,41].”

  • Lee MJ, Maliakal P, Chen L, et al. Pharmacokinetics of tea catechins after ingestion of green tea and (-)-epigallocatechin-3-gallate by humans: formation of different metabolites and individual variability. Cancer Epidemiol Biomarkers Prev. 2002;11(10 Pt 1):1025‐1032
  • Stalmach A, Troufflard S, Serafini M, Crozier A. Absorption, metabolism and excretion of Choladi green tea flavan-3-ols by humans. Mol Nutr Food Res. 2009;53 Suppl 1:S44‐S53. doi:10.1002/mnfr.200800169
  • Roowi S, Stalmach A, Mullen W, Lean ME, Edwards CA, Crozier A. Green tea flavan-3-ols: colonic degradation and urinary excretion of catabolites by humans. J Agric Food Chem. 2010;58(2):1296‐1304. doi:10.1021/jf9032975).

-In addition, I would recommend that the authors include in the discussion some data and prospects related to new delivery technologies able to improve the oral bioavailability of EGCG as viable alternatives for these specific samples.

As the referee suggests, discussion has been enlarge drawing attention to the use of these new delivery technologies in future clinical studies. The added paragraph is as follows (lines 499-505):

“New delivery technologies to improve the oral bioavailability of EGCG have raised to expand its application in a lipophilic media. Esterification of the water-soluble EGCG with stearic acid (SA), eicosapentaenoic acid (EPA), and docosahexaenoic acid (DHA) or replacing the hydroxyls of EGCG with acetyl groups have demonstrated a significant improvement in stability, bioavailability and bioactivities as antioxidant or antiproliferative [27,30]. For that their potential applications in future clinical studies should be analyzed in detail.”

Introduction has been enlarge with additional information about the topic of EGCG derivatives. We have emphasized that the esterification of EGCG solves the problem of its use in lipophilic environments, improving the spectrum of use in different matrices. The paragraph was added as follows (lines 106-114):

“Due to its hydrophilic nature, EGCG show homogeneity problems in lipid products, affecting not only its appearance but also its effectiveness. Structural modifications of EGCG via esterification with aliphatic molecules such as long-chain fatty acids have been performed to solve these problems and they can serve as a useful tool in altering its physical properties [27]. Interestingly, these ester derivatives have shown higher antiviral and antioxidant properties than the parent EGCG molecule and enhance its cellular absorption in vivo [27-29]. Moreover, EGCG derivates demonstrated to be more effective in neuroprotection that non-modified EGCG in a cellular model of Parkinson [30]. Thus, EGCG esters may be used in nutrition and cosmetics as lipophilic alternatives with no effects in its functional properties.”

References added:

(2011) Advances in Clinical Chemistry, 53, 155– 177

(2011) Journal of Agricultural Food Chemistry, 59, 6526–6533;

(2012) Journal of Functional Foods, 4, 87-93.

(2012) Food Chemistry, 131, 22–30.

(2010) Neuroscience Letters, 429, 36-364.